# A Python GPU-accelerated solver for the Gross-Pitaevskii equation and applications to many-body cavity QED

Lorenzo Fioroni[1,2†], Luca Gravina[1,2†], Justyna Stefaniak[1†], Alexander Baumgärtner[1], Fabian Finger[1], Davide Dreon[1★], Tobias Donner[1]

**1** Institute for Quantum Electronics & Quantum Center, Eidgenössische Technische Hochschule Zürich (ETHZ), Otto-Stern-Weg 1, CH-8093, Zürich, Switzerland
**2** Institute of Physics, École Polytechnique Fédérale de Lausanne (EPFL), CH-1015, Lausanne, Switzerland

★ dreond@phys.ethz.com
† These authors contributed equally

## Abstract

`TorchGPE` is a general-purpose Python package developed for solving the Gross-Pitaevskii equation (GPE). This solver is designed to integrate wave functions across a spectrum of linear and non-linear potentials. A distinctive aspect of `TorchGPE` is its modular approach, which allows the incorporation of arbitrary self-consistent and time-dependent potentials, e.g., those relevant in many-body cavity QED models. The package employs a symmetric split-step Fourier propagation method, effective in both real and imaginary time. In our work, we demonstrate a significant improvement in computational efficiency by leveraging GPU computing capabilities. With the integration of the latter technology, `TorchGPE` achieves a substantial speed-up with respect to conventional CPU-based methods, greatly expanding the scope and potential of research in this field.

## 1   Introduction

`TorchGPE` is a Python package specifically developed to numerically calculate the ground state and dynamical solutions of the Gross-Pitaevskii equation (GPE) in 2D. It provides extensive capabilities to handle advanced physical problems involving both linear potentials in the wave function $\psi$ and non-linear ones. The numerical methods employed in `TorchGPE` are based on the efficient split-step spectral algorithm, ensuring accurate and efficient computations. In comparison to various open-source packages solving the GPE, e.g `GPELab` [1, 2], `GPUE` [3], `spinor–GPE` [4], `BEC2HPC` [5], and others, the distinctive features of our work are:

- A user-friendly Python package with a modular approach to defining arbitrary linear, non-linear, time-dependent, and self-consistent potentials;

- A library of ready-to-use potentials of interest for the quantum gas community. This includes potentials modeling the dispersive matter-light interaction at the core of many-body cavity QED;

- Support for GPU optimization through the PyTorch library [6, 7], specifically targeting computational bottlenecks like fast Fourier transforms and Hadamard products.

The paper is organized as follows: in the next section, we introduce the Gross-Pitaevskii equation. Section 3 describes the algorithmic approach chosen. Following this, Section 4 outlines the GPU implementation and examines its performance on various devices and compared to the CPU-based implementation. Finally, in Section 5, we present benchmark results that provide a validation of our code. Despite our code being applicable to a vast class of problems, for the purpose of benchmarking we focus on Bose-Einstein condensates subject to optical potentials and possibly coupled to driven optical cavities.

## 2   Gross-Pitaevskii equation

The Gross-Pitaevskii equation is a nonlinear self-consistent partial differential equation that models the collective behavior of identical bosons in a condensate at effectively zero temperatures [8]. The time-dependent form of the GPE reads:

$$i\hbar\frac{\partial \psi}{\partial t} = \hat{H}_{\text{GPE}}(t)\psi = \left[-\frac{\hbar^2\nabla^2}{2m} + V_{\text{ext}}(\mathbf{r}, t, \psi) + g|\psi|^2\right]\psi, \tag{1}$$

where $\psi(\mathbf{r}, t)$ represents the condensate's wave function at position $\mathbf{r}$ and time $t$, $m$ denotes the mass of the bosonic particles, and $\hbar$ is the reduced Planck constant. This nonlinear equation incorporates a local potential $V_{\text{ext}}(\mathbf{r}, t, \psi)$, which may vary in time and space and self-consistently depend on $\psi$ itself. The term $g|\psi|^2$ describes the self-interaction among particles in a 3D condensate.

Solutions to the GPE are rarely analytical. Thus, efficiently obtained numerical solutions present a powerful tool to approximately compute many of the most relevant features of Bose-Einstein condensates such as density distributions, stability, dynamics, and collective

behaviors. Several computational methods have been developed to solve the GPE. Notable among these are Runge-Kutta integrators [9], Suzuki-Trotter solvers [10], the Crank-Nicolson method [11], and the split-step Fourier method [12]. Given the self-consistent nature of the GPE, these methods typically involve an iterative process. This process starts with an initial guess for $\psi$ and employs a contractive minimization approach [13] until convergence to a stationary configuration is reached (cf. Fig. 1). Among all methods, the split-step Fourier method is particularly efficient and compact, allowing both computation of the ground state and the real-time dynamics of the system [14].

## 3 Computational algorithm

### 3.1 Split-step Fourier method

The split-step Fourier method is a straightforward approach for evaluating the state $\psi(t)$ of a system at time $t$ by solving the evolution equation $i\hbar\partial_t\psi(t) = \hat{H}_{\mathrm{GPE}}\psi(t)$. The general solution to this non-linear Schrödinger equation can be written as

$$|\psi(t+\mathrm{d}t)\rangle = \mathcal{U}|\psi(t)\rangle = \exp\left(-\frac{i\hat{H}_{\mathrm{GPE}}(t)\mathrm{d}t}{\hbar}\right)|\psi(t)\rangle, \tag{2}$$

where $\mathcal{U}$ is the time-ordered evolution operator. To facilitate the computation, the Hamiltonian operator $\hat{H}_{\mathrm{GPE}} = \hat{T} + \hat{V}$ is divided into its kinetic energy component $\hat{T} = -\hbar^2\nabla^2/2m$, which is diagonal in reciprocal space, and the potential energy component $\hat{V} = V_{\mathrm{ext}}(\mathbf{r}, t, \psi) + g|\psi|^2$, which is diagonal in real space. By splitting the full evolution operator into $N$ steps corresponding to time intervals $\Delta t = t/N$, the evolution operator for a single time step takes the form

$$e^{-\frac{i}{\hbar}\hat{H}_{\mathrm{GPE}}\Delta t} = e^{\widetilde{T}/2}e^{\widetilde{V}}e^{\widetilde{T}/2}e^{\mathcal{O}(\Delta t^3)}, \tag{3}$$

where $\widetilde{T} = -\frac{i}{\hbar}\hat{T}\Delta t$ and $\widetilde{V} = -\frac{i}{\hbar}\hat{V}\Delta t$ are dimensionless operators. The computation of the evolution operators $\exp(\widetilde{V})$ and $\exp(\widetilde{T}/2)$ is performed in real and Fourier space, respectively. Denoting by $|\psi_k\rangle$ the wave function in Fourier space, the evolution over a single time step can be expressed as

$$|\psi_k(t+\Delta t)\rangle \approx e^{\widetilde{T}/2}\mathcal{F}\left[e^{\widetilde{V}}\mathcal{F}^{-1}\left[e^{\widetilde{T}/2}|\psi_k(t)\rangle\right]\right], \tag{4}$$

which is equivalent to the application of the full propagator up to third order in $\Delta t$. The Fourier transform operation, denoted as $\mathcal{F}$, is implemented using the Fast Fourier Transform (FFT) algorithm, applied to the wave function discretized over the computational grid. Notice that the Fourier transformation of the wave function automatically imposes periodic boundary conditions on the grid. The implementation of forward and backward Fourier transformations in the evaluation of the equation is associated with minimal computational expense compared to the direct calculation of the kinetic energy term by numerical differentiation. Note that due to its non-linearity, the potential has to be recalculated at each step, employing each time the most up-to-date wave function [15].

### 3.2 Adimensionalization of the GPE and contact interactions renormalization

In numerical computations, the challenge of handling operations on numbers with vastly different magnitudes is well-recognized. This issue, known as the *loss of significance* problem, arises due to the inherent limitations in floating-point precision of digital calculations. For example, when adding a very small number to a very large one, the contribution from the

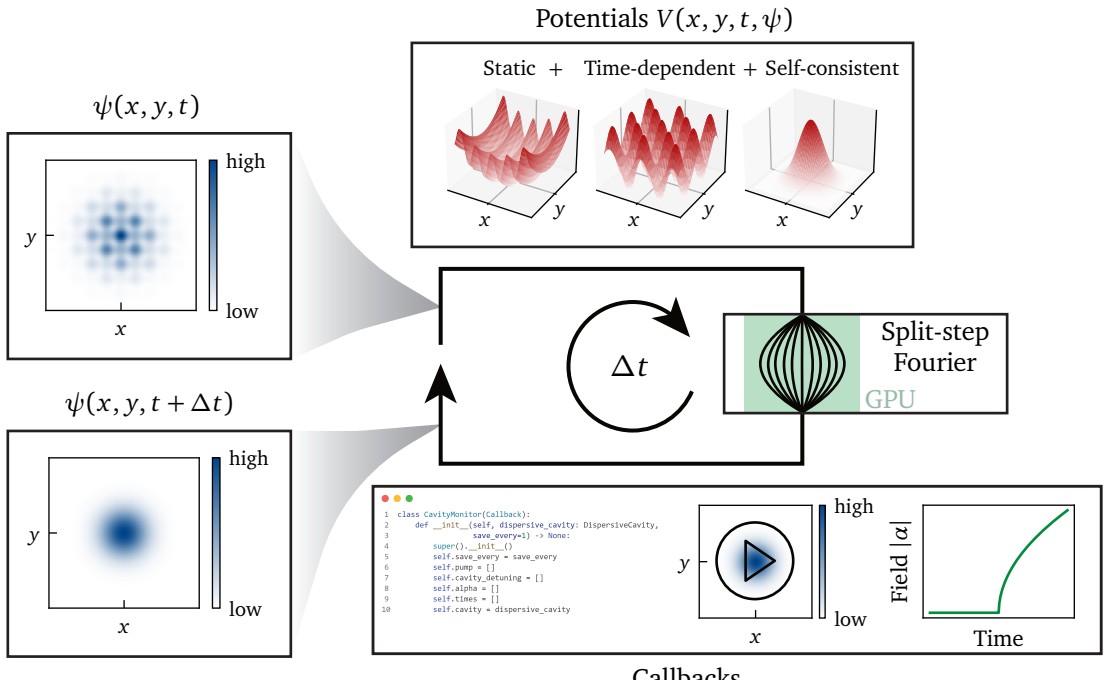

Figure 1: **Visualization of working principle**. The code starts with an initial wave function. Different types of potentials can be added in a modular fashion. The split-step Fourier method is implemented with PyTorch utilizing memory parallelization via GPUs. Callbacks allow the injection of custom code in the propagation, e.g., to read out observables during the evolution.

small number may be effectively lost due to the limited precision with which the numbers are stored.

This issue is most prominent when expressing the GPE in physical units [16]. To circumvent it, in TorchGPE, we reformulate the GPE using dimensionless variables, rescaled according to the system's natural units of length and time. Specifically, we employ the natural units of a quantum harmonic oscillator, with the frequency $\omega_\ell$ serving as the fundamental reference. This choice establishes the reference time and length scales, denoted by

$$\tau_\ell = \frac{1}{\omega_\ell} \quad \text{and} \quad \ell = \sqrt{\frac{\hbar}{m\omega_\ell}}, \tag{5}$$

respectively. Consequently, in our dimensionless framework, we define the time and length units as:

$$t' = \frac{t}{\tau_\ell} \quad \text{and} \quad \mathbf{r}' = \frac{\mathbf{r}}{\ell}. \tag{6}$$

Additionally, we rescale the energies and the wave function as

$$E' = \frac{E}{\hbar\omega_\ell}, \quad \text{and} \quad \psi' = \ell^{d/2}\psi, \tag{7}$$

where $d$ is the dimension of the system. This rescaling ensures that our numerical computations are stable and accurate without altering the problem's underlying physics.

A complete description of a physical system usually requires performing simulations in a tridimensional space. While our code is, in principle, able to perform such simulations, it is true that often the relevant physics can be extrapolated from effective descriptions in lower

dimensional spaces where simulations are much less computationally challenging. This occurs, for instance, in several experimentally accessible devices with a strong harmonic confinement in one or two directions. If the dynamics unfolds primarily over the $xy$ plane, equation (1) is virtually unchanged, except for the contact interaction strength which is rescaled as

$$g_{2D} = \frac{g}{\sqrt{2\pi}a_\perp}.\tag{8}$$

The renormalization length $a_\perp$ allows to approximate the physics of the confined 3D gas through an effective 2D description [17, 18]. Note that in the weakly interacting limit, $a_\perp$ is well approximated by the harmonic oscillator length along the transverse direction (cf. equation (5)).

### 3.3 Imaginary-time evolution

The imaginary-time evolution algorithm is a commonly used method to find the minimal energy solution $|\psi_0\rangle$ of the GPE [11, 19–22]. The method is based on a transformation to imaginary time ($t \to -i\tau$), often referred to as Wick rotation, leading to the exponential decay of all states relative to the ground state. The underlying principle of this method parallels that of the power method and ensures that in the limit of $\tau \to \infty$, the evolved wave function ultimately converges to the ground state of the system $|\psi(\tau \to \infty)\rangle \approx |\psi_0\rangle$. This outcome holds independently of the initial wave function chosen, as long as this is not orthogonal to $|\psi_0\rangle$.

The problem of determining the ground state of the system can thus be addressed by evolving an arbitrary initial wave function for a sufficiently long imaginary duration. In TorchGPE, this evolution is achieved using the split-step Fourier propagation method discussed above, where the time step $\Delta t$ is replaced with $-i\Delta\tau$. Due to the diffusive nature of the resulting equation, the wave function must be normalized after each iteration to prevent it from decaying completely.

## 4   GPU-accelerated PyTorch implementation

Originally designed for 3D graphics rendering, Graphical Processing Units (GPUs) have become pivotal in computationally intensive domains like Deep Learning. In this study, we harness the GPU's robust data parallelization capabilities to enhance the efficiency of the split-step spectral method. The FFT algorithm and Hadamard products are particularly well-suited for the exploitation of the parallel processing capabilities of GPUs. Drawing on the insights of Ref. [4], our approach utilizes the CUDA programming framework in conjunction with the PyTorch library. This combination enables efficient processing of large datasets, markedly surpassing CPU performance in tasks such as matrix-matrix multiplications and Fourier transformations. TorchGPE's default behavior is to execute on GPUs when a compatible NVIDIA processor is present; otherwise, it resorts to using the CPU.

This section offers a comparative analysis of our code's performance across various CPUs and GPUs. Our focus is to assess the effectiveness of GPU integration in boosting computational efficiency, highlighting the advantages of this approach in handling complex, data-intensive operations.

### 4.1 Computational performance

To accurately compare GPU and CPU performance, we adopt the methodology from Ref. [4] and, to minimize the impact of slow data transfers between CPU RAM and GPU memory, we pre-allocate the necessary tensors on the respective devices. The figure of merit of our

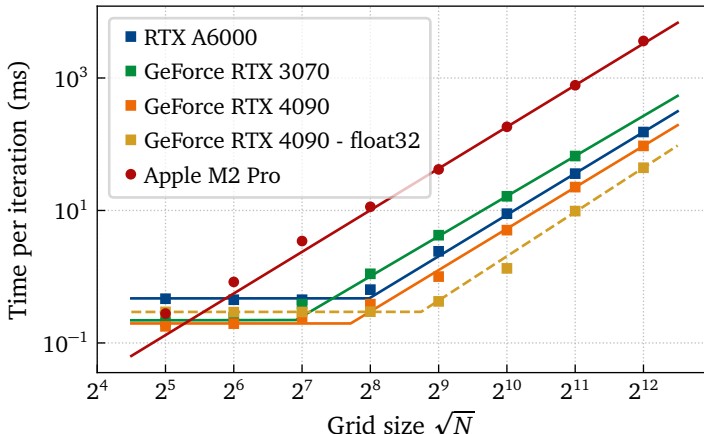

Figure 2: **Benchmark**. Performance comparison between the implementation of the imaginary time propagation algorithm for different grid sizes $N_x = N_y = 2^n$ on a CPU (circles) and several NVIDIA GPU processors. The simulated system is a trapped, non-interacting BEC, with the only potential being a fixed harmonic confinement. The data represented via the dashed line have been obtained by storing numerical values in the single-precision floating-point format, as opposed to the double-precision format used for the other simulations.

benchmark is the execution time averaged over five independent runs, each involving $10^3$ steps of imaginary time propagation, across various two-dimensional grid sizes ($N = N_x \times N_y$). The results, depicted in Fig. 2, highlight the performance differences and the significant speedup achieved with GPU acceleration.

The data reveals a consistent and substantial speedup in GPU execution for all grid sizes and processors tested. Notably, the speed-up becomes significant for $\sqrt{N} \gtrsim 2^6$ points per side, reaching up to a **40**-fold increase for $\sqrt{N} \gtrsim 2^9$. Note that the comparison is conducted on grids where the number of samples in each direction is an integer power of two, where the FFT algorithm is maximally efficient [23]. Nevertheless, we do not record qualitative deviations from the observed trend for non-optimal grid sizes. The GPU scaling behavior presents two distinct patterns. For smaller grids, the execution time appears constant. Conversely, for larger grids, the behavior mirrors that of CPUs, showing a power-law dependence on $N$. This shift is interpreted as the GPU reaching its limit for simultaneous data operations, necessitating sequential batch processing for larger grids. Additionally, using regular precision floats, which require half the memory, we can extend the range of values of $N$ where this efficient flat computation rate is observed.

# 5 Benchmarking

A benchmark on a physical system is essential to assess the performance and accuracy of our code. It allows us to validate the computational model against known experimental or theoretical results, ensuring its reliability or identifying potential discrepancies.

## 5.1 Simulating Kapitza-Dirac diffraction of a BEC in an optical lattice

As a first example, we explore the diffraction of a BEC from an optical lattice switched on for a time $\tau$. We consider an optical lattice created by two counterpropagating lasers of wave-

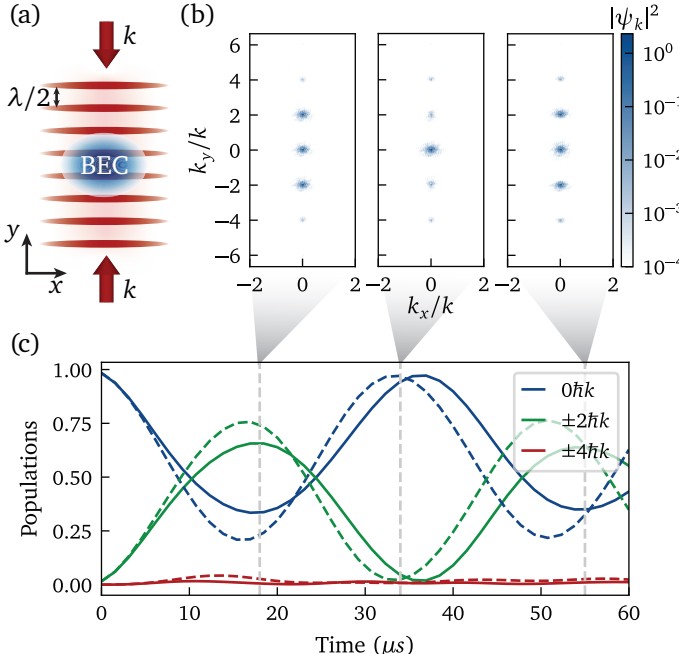

Figure 3: **Kapitza-Dirac diffraction of a trapped BEC**. (a) Experimental configuration. A trapped BEC is illuminated by two counterpropagating Gaussian beams described by quasi-plane waves with wavevector $k$, inducing a standing-wave modulation along the $y$-direction with period $\lambda/2$. (b) Representative momentum-space probability distributions for $t = (18, 34, 55)\,\mu s$. (c) Time evolution of different momentum mode populations. The solid (dashed) curves are calculated for an $s$-wave scattering length of $a_s = 300\,a_B$ ($a_s = 0\,a_B$).

length $\lambda$ along the $y$-direction. They interfere and form a standing wave with electric field $E(y) = E_0 \Pi(t/\tau) \cos(ky) e^{-i\omega_p t}$, of amplitude $E_0$, wavenumber $k = 2\pi/\lambda$ and angular frequency $\omega_p$ (cf. Fig. 3a). $\Pi$ is a unit pulse of duration $\tau$, corresponding to the time that the lasers are turned on. The corresponding potential energy in Eq. (1) is

$$V(x, y, t) = V_0(t) \cos^2(ky) + V_t(x, y), \tag{9}$$

where $V_0 = -\alpha(\lambda)|E_0|^2 \Pi(t/\tau)$ is the light shift experienced by the atoms, and $\alpha(\lambda)$ the scalar atomic polarisability [24]. In the potential, we also consider an additional (isotropic) harmonic confinement for the BEC $V_t = \frac{1}{2}m\omega_t^2(x^2 + y^2)$ with the trapping frequency $\omega_t$.

Diffraction from a lattice was one of the first applications of coherent atom optics [25, 26]. Nowadays, this method is commonly used to measure the depth of an optical lattice since it can be modeled by solving the Schrödinger equation of motion of the lowest few momentum states $|2n\hbar k\rangle$, which is accurate at low interaction strengths $g$ and short times $\tau$. The experimental protocol consists of flashing the lattice for a duration $\tau$ and recording the population in the different momentum states after ballistic expansion. By repeating the experiment for varying pulse durations $\tau$, the lattice depth can be determined. While in experiments, the momentum-space populations are typically retrieved from time-of-flight measurements, in our simulations, we can directly access them, at no additional cost, from the wave-function amplitude in reciprocal space (cf. Fig. 3b). In Fig. 3c, we plot the results of the simulation for both the interacting and the non-interacting cases, showcasing the coherent oscillations of the diffracted populations.

## 5.2 Numerical calculation of the self-organization phase diagram of a BEC in an optical cavity

Self-organization refers to the spontaneous formation of ordered patterns in the condensate, driven by the balance between kinetic energy and cavity-mediated atomic interactions [27]. These interactions give rise to spatial patterns characterized by symmetric density distributions. The emergence of such patterns in a BEC breaks two continuous symmetries: phase invariance for superfluidity and translational invariance for crystal formation. In this context, the lattice structure emerging from a self-organized BEC is also known as *lattice supersolid*, a quantum phase that intriguingly combines crystallization in a many-body system with the dissipationless flow typical of superfluids [27]. Given the self-consistent nature of this problem, our framework allows for an efficient simulation of the physics.

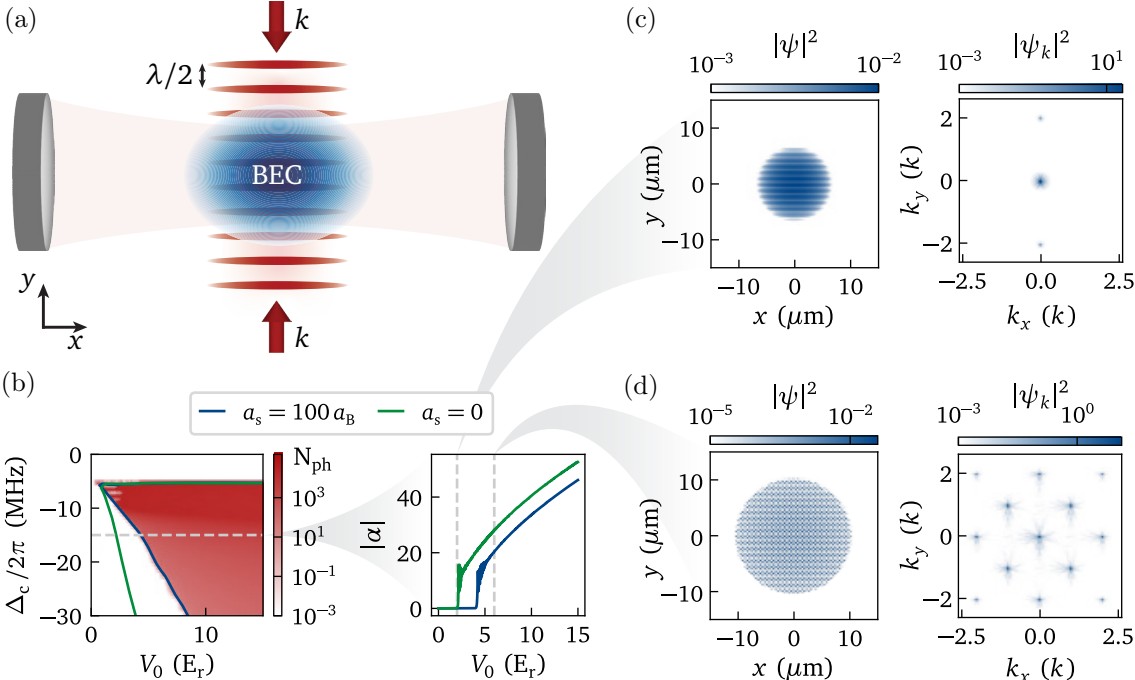

Figure 4: **Self-organization of a trapped BEC in an optical cavity**. (a) Experimental configuration. Two counter-propagating Gaussian beams illuminate a trapped BEC, forming a standing wave modulation along the $y$ direction. The BEC is placed in the center of an optical cavity with its axis along $x$. (b) (Left) Mean number of photons $N_{ph}$ in the cavity for different values of pump strength $V_0$ and cavity detuning $\Delta_c$. The pump strength is expressed in units of the recoil energy $E_r = (\hbar k)^2/2m$. In blue (green) the phase boundary for an $s$-wave scattering length of $a_s = 100\, a_B$ ($a_s = 0\, a_B$) is shown. (Right) Corresponding values of $\alpha$ in a longitudinal cut through the phase diagram detailing the gradual development of a non-vanishing order parameter characteristic of a second-order phase transition. The pump power is linearly increased from $0$ to $15\, E_r$ in a total time of $15\, ms$. (c) Wave function in real (left) and momentum space (right) of the interacting BEC in the normal phase. The cavity detuning has been set to $\Delta_c = -2\pi \cdot 15$ MHz and the pump strength to $V_0 = 2\, E_r$. (d) Wave function in real (left) and momentum space (right) of the interacting BEC in the organized phase. The cavity detuning has been set to $\Delta_c = -2\pi \cdot 15$ MHz and the pump strength to $V_0 = 6\, E_r$.

In this section, we use our code to numerically solve the GPE associated with this model, characterizing the physics observed in the experimental realizations of the latter [28]. In the mean-field limit, the potential energy is

$$V(\mathbf{r}) = V_t(\mathbf{r}) + V_p(\mathbf{r}) + V_c(\mathbf{r}) + V_i(\mathbf{r}), \tag{10}$$

where $V_t$ is the trapping potential and $V_p$, $V_c$, and $V_i$ denote the pump, cavity, and interference lattices, respectively. The trapping potential is harmonic and time-independent, while the others are defined as:

$$V_p = V_0 \cos^2(ky), \tag{11}$$

$$V_c = U_0 |\alpha|^2 \cos^2(kx), \tag{12}$$

$$V_i = 2\sqrt{V_0 U_0}, \mathcal{R}e(\alpha) \cos(kx) \cos(ky), \tag{13}$$

where $V_0$ is the depth of the pump lattice, $U_0$ the depth of the single-photon lattice inside the cavity, and $\alpha$ the coherent field amplitude in the cavity. Notably,

$$\alpha = N\sqrt{V_0 U_0} \frac{\int \cos(kx)\cos(ky)|\psi|^2 dx\, dy}{\tilde{\Delta}_c + i\kappa} \tag{14}$$

is explicitly dependent on the atomic wave function at time $t$, introducing an additional element of self-consistency to the problem. As a result, $\alpha$ is calculated at each time step using the current value of the density $|\psi(t)|^2$. The term appearing at the denominator

$$\tilde{\Delta}_c = \Delta_c - U_0 \int \cos^2(kx)|\psi(x,y)|^2 dx\, dy \tag{15}$$

is the dispersively shifted cavity detuning. A formal justification of the self-consistent potential is beyond the scope of this work. For further details, we refer the reader to Ref. [27].

In Fig. 4, we present results that stem from the solution of the GPE in both imaginary- and real-time. Replicating the experimental setting shown in Fig. 4a, where a BEC is illuminated by a laser orthogonal to the cavity mode, we observe the presence of two different phases. For values of $V_0$ smaller than a critical threshold, the density distribution is only modulated by the transverse lattice (cf. Fig. 4c) and the cavity population is vanishing (cf. Fig. 4b). Conversely, as $V_0$ is increased beyond its critical value, the system self-organizes into the chequerboard density modulated phase displayed in Fig 4d. Concurrently, the cavity acquires a non-vanishing population as seen in Fig. 4b. We display simulations for both interacting and non-interacting gases, revealing that repulsive interatomic interactions in the BEC increase the critical pump power necessary for the system to self-organize. Not captured by low-energy theories, this behavior is well-reproduced in our GPE simulations and visible in Fig. 4b.

This implementation can also be generalized to explore previous experiments on multi-cavity systems or on setups with a running-wave component added to the transverse beam, where different symmetries appear [29–31] and new dynamical effects emerge [32].

## 5.3 Simulated roton spectroscopy at the self-organization phase transition

Fundamental insight into the underpinnings of emergent many-body effects in quantum fluids is provided by the investigation of the spectrum of their elementary excitations. Indeed, the self-organization phase transition can be understood as resulting from a virtual process where cavity photons mediate an effective long-range interaction between the atoms. Long-range interactions have long been predicted to give rise to a softening of the excitation spectrum at a finite momentum $\mathbf{k} = \mathbf{k_{rot}}$, similar to that observed in superfluid helium [33] and, more

recently, in dipolar BECs [34, 35]. Under simplifying assumptions, e.g. infinite uniform gas, the correction to the free-particle dispersion relation $\epsilon_k$ can be derived in the framework of the Bogolyubov theory [36, 37]. This results in the dispersion relation

$$\tilde{\epsilon}_k = \epsilon_k \sqrt{1 + \frac{2g_{2D}|\psi(r)|^2 + 4N\mathcal{V}\tilde{V}(k)}{\epsilon_k}}, \tag{16}$$

where $\epsilon_k$ is the free kinetic energy, $\tilde{V}(k)$ is the Fourier transform of the mode structure of the long-ranged potential $V(r) = \cos(\mathbf{k}_c \cdot \mathbf{r})\cos(\mathbf{k}_p \cdot \mathbf{r})$, and $\mathcal{V} = \hbar V_0 U_0 \tilde{\Delta}_c / (\tilde{\Delta}_c^2 + \kappa^2)$ is its effective strength. The presence of the cavity-mediated long-range interaction leads to a renormalization of the energy spectrum around the wavevector $k$, thus creating a minimum of energy at finite momentum, like illustrated in 5b.

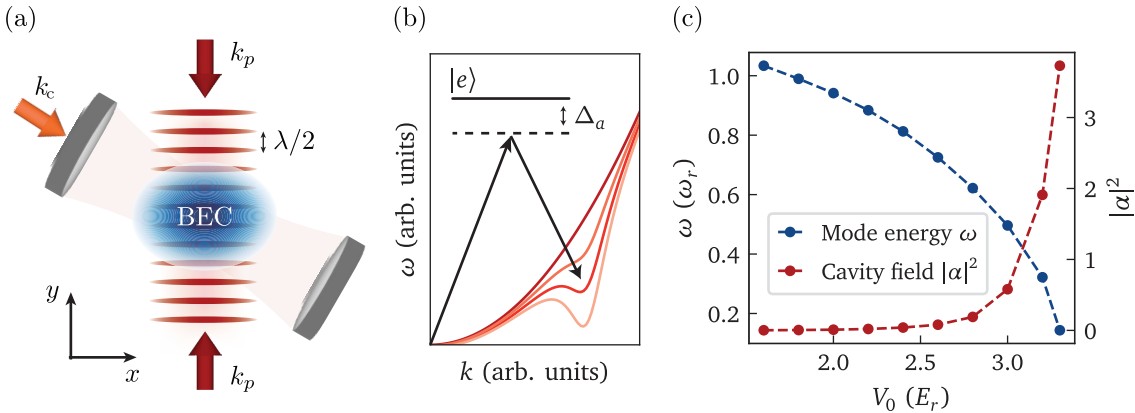

Figure 5: **Bragg spectroscopy of Soft Mode in Lattice Supersolid.** (a) Schematic of the simulated system. Atoms in the optical cavity with the applied transverse pump lattice ($k_p$) and the applied Bragg probe beam ($k_c$). (b) Sketch of the mode softening in the dispersion relation. The arrows indicate the role of the pump and probe beams, where the pump is detuned by $\Delta_a$ from the excited atomic state $|e\rangle$. (c) Mode softening (blue, left axis) and cavity field (red, right axis) extracted from the GPE simulation as the transverse pump power $V_0$ approaches its critical value. The gas is made of $N = 2 \cdot 10^5$ $Rb^{87}$ atoms trapped in a harmonic potential with frequency $\omega = 2\pi \cdot 100\,\text{Hz}$. The cavity detuning has been set to $\Delta_c = -2\pi \cdot 20\,\text{MHz}$, and the atomic one to $\Delta_a = -2\pi \cdot 76.6\,\text{GHz}$.

Experimentally, the roton spectrum can be probed through a variant of Bragg spectroscopy [38]. After preparing the system at a given interaction strength, the cavity field is excited with a weak pulse along the cavity axis, with frequency detuned by $\Delta_p$ from the frequency of the driving laser. In the regime of adiabatic elimination of the light field dynamics, the on-cavity axis probe is modeled by modifying the cavity field in equation (14) to $\alpha'(t) = \alpha + \delta\alpha(t)$ with

$$\delta\alpha(t) = \frac{\Omega_c \exp(i(\Delta_p t + \phi))}{\Delta_c - U_0 \int \cos^2(k_p r)|\psi(r)|^2 d^2r + i\kappa}. \tag{17}$$

Here, $\Omega_c$ is the strength of the on-cavity axis driving field, and $\phi$ is the phase of the probe beam. The interference between the cavity probe and the transverse pump results in an amplitude-modulated probing potential

$$V = 2\sqrt{V_0 U_0} Re(\alpha) \cos(k_p r) \cos(k_c r), \tag{18}$$

Such modulation acts on the BEC and results in photons being scattered from the pump to the cavity field, and hence in the oscillation of the latter. Therefore, the light leaking out of the cavity follows the oscillatory evolution of the density modulation, and the system's response to the modulation can be observed in the intracavity photon number. The excitation energy can be extracted by monitoring the field leaking out of the cavity and analyzing the response of the system to different probing frequencies.

In Fig. 5, we present the results of the cavity spectroscopy protocol performed with `TorchGPE`, showing the mode softening as the phase transition is approached. In Fig. 5a we sketch the experimental apparatus, and in Fig. 5b the qualitative behavior of rotonic excitations. Fig. 5c shows numerical results of the excitation energy of the soft mode accompanied by the diverging response in the cavity light field. The results have been obtained by first computing the ground state of the gas via imaginary time propagation and subsequently evolving the system in real-time while probing for $4$ms with an amplitude of $\Omega_c = 100 E_r$. The mean photon number in the cavity has been computed for different values of the detuning between the probe and the pump $\Delta_p$, and fitted with a Gaussian function [38]. The frequency of the soft mode and the strength of the response have been extracted from the fit results. Bragg spectroscopy also allows determining the dynamic structure factor of the system around the mode softening, providing information about the emergence of quasi-particle modes and their energy [39]. This can be computed with `TorchGPE`, as it provides direct access to both the cavity field and the condensate's wave function.

## 6 Outlook

The future prospects of the presented package include three main areas of development. First, extending the method to three dimensions would broaden its applicability to a wider range of physical systems. Second, the use of an unevenly spaced computational grid would allow the sampling to be adjusted to the characteristics of the potentials. Consequently, it would reduce the integration time without greatly affecting the quality of the results. Lastly, incorporating spinor wavefunctions would enable a description of systems involving spin degrees of freedom. These future directions aim to enhance the package's versatility, accuracy, and usability.

## Acknowledgement

We are grateful to Tilman Esslinger for fruitful discussions. We acknowledge funding from the Swiss National Science Foundation (Project No. IZBRZ2 186312) and from the Swiss State Secretariat for Education, Research and Innovation (SERI).

**Author contributions**   LG extended and optimised the initial versions of the code and integrated the GPU's capabilities. LF developed the current version of the package and the public documentation. JS expanded the scope of the code and supervised the final development. FF and AB provided input on the physical models and technical topics. DD conceived the project, TD and DD supervised its development. All authors contributed to the manuscript.

**Code availability**   `TorchGPE` is openly available at the following repository: https://github.com/qo-eth/TorchGPE, along with documentation and examples.

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
