# Peer review of "A Python GPU-accelerated solver for the Gross-Pitaevskii equation and applications to many-body cavity QED"

_SciPost Physics Codebases, doi:SciPost Phys. Codebases 38 (2024) , SciPost Phys. Codebases 38-r1.0 (2024)_

## Round 1 · Referee Report · Anonymous (Referee 1) · 2024-5-23

Strengths

1) Clearly written library that should be easy to use 2) Good documentation of the features that are available

Weaknesses

1) Lack of comparison to other available libraries 2) Quite a few features missing which would make this a really useful toolbox

Report

The paper by Fiorioni et al introduces a new GPU accelerated python library for simulating the GPE in a range of potentials. The documentation included with the library is clear and the examples chosen and described are simple enough to easily follow but complicated enough to show a range of capabilities of the software. The structure and design of the software package should make it useful for researchers working in this area, especially when the additions mentioned in the final section are included. The paper would be significantly improved with the addition of more performance benchmarks for the more complicated models studied and also comparison to other libraries which are available for finding solutions to the same equations.

Requested changes

1) For each of the examples studied the authors should present a study of how well their algorithm converges to the correct result with eg the size of the grid, along with the required computing resources.

2) It would also be useful to add some comparisons to other libraries. How much more efficient is the code here?

3) The link at the end of the paper does not have the correct address

Recommendation

Ask for minor revision

  • validity: high
  • significance: ok
  • originality: ok
  • clarity: top
  • formatting: excellent
  • grammar: perfect

Author:  Davide Dreon  on 2024-09-01  [id 4729]

(in reply to Report 1 on 2024-05-23)

Dear Referee,

We thank you for carefully reading our manuscript and for providing very useful comments and suggestions for improving it. We agree that adding a more in-depth benchmarking of the code adds to the value of our work and that a direct comparison to other libraries can be useful. This is also in line with the suggestions of Referee 2 who asked for a benchmarking of the code against analytical or other known results.

We have provided a detailed response with some informative graphics in the pdf attached to this response.

We believe to have addressed all of your requested changes in the new version of the manuscript and hope that you will support its publication.

The authors

Attachment:

referee_1_reply.pdf

---

## Round 1 · Referee Report · Anonymous (Referee 2) · 2024-7-17

Strengths

1- The package provides easy access to powerful GPU-based methods. 2- The package is flexible enough to simulate a large variety of problems. 3- The paper is written in a clear and fairly pedagogical format.

Weaknesses

1- Lack of quantitative comparisons to known analytic results. 2- Lack of comparison or analysis between the tested GPU models.

Report

The paper by Fioroni et al. showcases the versatility and speed of their new GPU-enabled GPE solver. The paper is pedagogically written and easy to follow, while also demonstrating the variety of problems that may be approached. The advantage of using a GPU is demonstrated in a clear way. Overall, the paper, software, and documentation could prove to be a valuable tool for the ultracold gas and quantum optics communities.

Our main criticism is a lack of quantitative verification of the numerical methods against known analytic results in the paper. The paper does a great job showcasing the versatility of the software in many fairly complicated settings. This is great, but the results are only verified qualitatively: the Kapitza-Dirac oscillations of Figure 3, the phase diagram and BEC density modulation in Figure 4, and the roton softening in Figure 5 all look correct, but no quantitative analysis of these results seems to be made. Analytic results are not always possible of course, necessitating tools like this software. However, before going beyond known analytic results it is good to benchmark against those available. A few suggestions for quantitative comparisons are listed in the "requested changes" section below.

It is clear that the GPUs outperform the CPU. However, the differences in the GPU models was not discussed in the paper. It would be useful to include a discussion of any significant differences in the hardware specifications of the tested GPUs and whether any of those differences impacted performance.

In summary, we recommend the paper for acceptance after some quantitative checks of the numerical methods are demonstrated.

Requested changes

  • The Kapitza-Dirac results in Figure 3c would benefit from a quantitative comparison to the standard method of integrating the Schrodinger equation in momentum space, as alluded to in the main text of the paper. This should show agreement with the simulation results for the a = 0 simulations at early times. This would provide both a quantitative check for the numerical methods, and show in what ways the simple method of integrating the Schrodinger equation breaks down at later times and for larger interaction strengths.

  • It could be a good addition to include simulations of just a BEC in a harmonic trap before considering more complicated scenarios. This would allow one to see convergence to the analytic Thomas-Fermi distribution as one increases the atom number, for example.

  • Please specify what $\kappa$ is (presumably the cavity loss rate) starting in Eq. 14.

  • Please specify the atom number $N$ used for the simulations in Figure 4 so that the photon counts may be verified.

-Please comment on the differences between the tested GPU models. What is different about the hardware of these GPUs? Did the hardware specifications make any significant differences in performance, possibly beyond what is seen in the Figure 2 comparison to the CPU?

Recommendation

Ask for minor revision

  • validity: high
  • significance: high
  • originality: good
  • clarity: top
  • formatting: excellent
  • grammar: perfect

Author:  Davide Dreon  on 2024-09-01  [id 4730]

(in reply to Report 2 on 2024-07-17)

Dear Referee,

We appreciate your careful review of our manuscript and the valuable comments and suggestions provided for its improvement. We are pleased to learn that you recommend acceptance of our manuscript after we presented quantitative validations of our numerical methods. This aligns also with the suggestion of Referee 1 to add a benchmarking of the algorithm by demonstrating its convergence.

In the pdf attached to this response, we have provided a point-by-point response with some informative graphics.

We believe that all of the requested changes have been adequately addressed in the updated version of the manuscript and hope that you will provide your support for the publication of the revised version.

The authors

Attachment:

referee_2_reply.pdf

---

## Round 3 · Referee Report · Anonymous (Referee 1) · 2024-9-13

Strengths

See previous report

Weaknesses

See previous report

Report

The authors have addressed both my and the other referees comments on their manscript fully. It is is now suitable for publication.

Requested changes

None

Recommendation

Publish (meets expectations and criteria for this Journal)

---

## Round 3 · Referee Report · Anonymous (Referee 2) · 2024-10-1

Report

The revised manuscript has satisfactorily addressed the points raised in the report. I thank the authors for their reply and recommend the revised manuscript for acceptance.

Recommendation

Publish (easily meets expectations and criteria for this Journal; among top 50%)

---

## Round 3 · Author Response

Dear Editor, dear Referees,

Thank you for your positive assessment of the paper and for providing us with very constructive and useful reports.

We are pleased to read that both Referees liked the pedagogical style of the paper and the variety of problems our code could help solve with its GPU speedup. The Referees also agreed on the usefulness of additional benchmarks. We have followed their suggestions and added a use case, while more examples and details of how to use the package have been kept in the software documentation to make the paper easier to read.

We have already included a point-by-point response to each of the referees' reports. In the "list of changes" below is a summary of the main changes made to the paper.

We believe that we have addressed all their recommendations and we hope that this version is suitable for publication in SciPost Physics Codebases.

With kind regards,
Davide Dreon on behalf of all authors of the manuscript

---

## Round 3 · List of Changes

• In Section 4.1, highlighted the key characteristics of a GPU that most significantly affect the performance of the code
  • Added a comparison with another library in Section 4.1, to highlight the performance improvements. Due to its popularity, we have chosen to compare our library to GPELab
  • Added the additional benchmark of a harmonically trapped BEC (section 5.1)
  • Added comments on benchmarks convergence
  • Reference to our previous experimental work where a preliminary version of TorchGPE had been used
  • Minor corrections and specifications (particle numbers used, grid sizes, kappa definition) as suggested by the Referees

---

## Editorial Decision

published